Journal of Machine Learning Research 1 (2021) 1-48          Submitted 07/21; Published XX/XX

# SparseConvMIL: Sparse Convolutional Context-Aware Multiple Instance Learning for Whole Slide Image Classification

**Anonymous**

**Editor:**

## Abstract

Multiple instance learning (MIL) is the preferred approach for whole slide image classification. However, most MIL approaches do not exploit the interdependencies of tiles extracted from a whole slide image, which could provide valuable cues for classification. This paper presents a novel MIL approach that exploits the spatial relationship of tiles for classifying whole slide images. To do so, a sparse map is built from tiles embeddings, and is then classified by a sparse-input CNN. It obtained state-of-the-art performance over popular MIL approaches on the classification of cancer subtype involving 10000 whole slide images. Our results suggest that the proposed approach might (i) improve the representation learning of instances and (ii) exploit the context of instance embeddings to enhance the classification performance. The code of this work is open-source at *github censored for review*.

**Keywords:** Multiple Instance Learning, Whole Slide Image Classification, Large-scale Histopathology

## 1. Introduction

An extremely large number of histological routine tasks involve the classification of whole slide images, including subtype diagnostic, tumor screening, tumor grading, or the choice of treatment. However, the extreme sizes of whole slide images impede their classification with conventional deep learning architectures which are the gold standard for classification in medical images (Litjens et al., 2017). Indeed, while traditional image weighs less than 1 megapixel — *e.g.* 0.09 megapixel for images of ImageNet (Deng et al., 2009) — whole slide images often contain several billions of pixels at full magnification. Unfortunately, classical deep learning architectures are not suited for such large images due to memory issues. For instance, ResNet200 (He et al., 2016) can only fit 32 images of width 224 for simultaneous forward and backward pass on popular graphic cards — equivalent to only around 1.6 megapixels.

WSI classification is a challenging problem. It cannot be tackled by downsampling WSI because many tasks rely on the phenotype of cells which is lost with downsampling. Classical approaches extract handwritten features from annotated elements of interest such as tumor tissue or lymphocytes and use traditional machine learning (Beck et al., 2011; Wang et al., 2014; Saltz et al., 2018). Apart from the limited power of manually designed features, this approach is impeded by the many difficulties of obtaining accurate annotations although recent approaches aim at automating the delineation of elements of interest (Saltz et al., 2018; Lerousseau et al., 2021). Meanwhile, classifying a (randomly) subsampled contiguous

region from a WSI may result in a non-representativeness of a WSI, a phenomenon known as tumor heterogeneity (Heppner and Miller, 1983; Marusyk and Polyak, 2010). To circumvent tumor heterogeneity and the limitations of graphic cards memory, a solution consists in classifying a (pseudo-)uniformly sampled set of tiles using multiple instance learning (MIL) (Keeler et al., 1991; Dietterich et al., 1997). However, the majority of MIL approaches do not consider the relationship of tiles which undoubtedly yield valuable information. To the best of our knowledge, the only viable spatially-aware WSI classifying solution is Streaming CNN (Pinckaers et al., 2019). This approach leverages the divide and conquer paradigm by splitting a gigapixel image in subimages which are sequentially processed by graphic cards until the signal can fit wholly on video memory. While this approach can effectively entirely process gigapixel images, it suffers from additional processing time and memory usage due to the intermediate storage of temporary forward and backward maps.

The objective of our work is to bridge the gap between traditional image classification and multiple instance learning for whole slide images. The purpose is to enhance WSI classification and tile representation learning with a scalable and modular tool. To this end, we propose a fully differentiable context-aware multiple instance learning paradigm that exploits the spatial relationship of tiles extracted from whole slide images. To do so, a sparse map is built by mapping the tiles embeddings to the locations of their associated tiles within the original WSI. Then, a sparse-input CNN computes a WSI embedding from the sparse map, which is further classified using a generic classifier. The potentials of this approach is benchmarked on (i) a traditional histological MIL task, and (ii) an original large-scale experiment involving 10000 whole slide images from The Cancer Genome Atlas for subtype classification among 32 classes. Our contributions are twofold:

- a modular and powerful multiple instance learning framework

- a very large scale experiment involving 10000 slides on a task extremely pertinent to cancer clinical histopathological routine

## 2. Background

### 2.1 Multiple instance learning

MIL is a particular classification paradigm where the considered objects are called bags (here, WSI) and are made of other objects called instances (here, patches or tiles). Instances may or may not have labels, although in any case those are unavailable during training. The only available information is the label of bags. In the more general case, MIL models can be mathematically decomposed into 3 parts: (i) an instance embedder $f_{\theta_1}$ that converts each instance into an embedding, (ii) a pooling operator $g_{\theta_2}$ that computes a bag embedding from the instance embeddings, and (iii) a generic classifier $h_{\theta_3}$ that converts the bag embedding into scores, such that a bag $(x_1, \ldots, x_K)$ is predicted with

$$h_{\theta_3}\Big(g_{\theta_2}\big(f_{\theta_1}(x_1), \ldots, f_{\theta_1}(x_K)\big)\Big) \tag{1}$$

$f_{\theta_1}$ can be any type of embedding function, with or without parameters, differentiable or not. In particular, instance-based MIL is a special case where $f_{\theta_1}$ outputs embeddings that

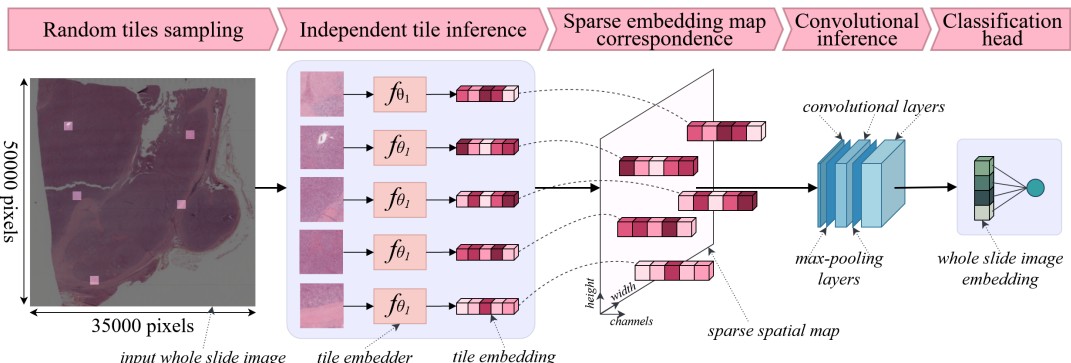

Figure 1: Visual representation of the proposed approach. First, a set of patches are randomly sampled throughout a WSI (here, 5 patches), and are then concurrently and independently forwarded into a shared patch embedder $f_{\theta_1}$. Then, a sparse map is built by placing each resulting embedding at the location of its associated patch. This map is forwarded into a sparse-input CNN producing a bag embedding which is finally classified into scores or probabilities using a generic classifier $h_{\theta_3}$.

lie in a unidimensional space, *i.e.* apparent to a probability space. The classifier $h_{\theta_3}$ can be any type of classifier, including a multi-layer perceptron. Actually, most of the MIL community efforts have revolved around pooling operators which can be grouped in two categories: permutation invariant operators, and others.

In some applications, such as the drug discovery problem (Dietterich et al., 1997), instances do not exhibit dependency, ordering, or spatial information among each other, *i.e.* they are independently and identically distributed (iid). For any permutation $\sigma$, the output of an iid pooling operator is the same for $x = (x_1, \ldots, x_K)$ and $(\sigma(x_1), \ldots, \sigma(x_K))$, *i.e.* they are permutation invariant. Examples of such pooling functions are max, mean, log-sum-exp (Ramon and De Raedt, 2000), attention-based (Ilse et al., 2018) — their mathematical formulations are provided in Appendix A.

The iid assumption does not hold for applications where there is inherent structural information about instances, such as document classification from sentences (Angelidis and Lapata, 2018), or WSI classification from tiles. Zhou et al. (2009) have notably achieved state-of-the-art performance over iid MIL pooling operators by building a graph from instance embeddings, and then performing classification with kernel methods. With recent improvements of graph neural networks (Wu et al., 2020), further iterations have been proposed by the community (Tu et al., 2019; Yi and Lin, 2016; Zhao et al., 2020).

## 2.2 Sparse-input convolutional neural network

Sparse data involve the concept of *active* and *inactive* cells or pixels, where inactive cells contain no data — not even a value of 0 which uses memory. Truly sparse data have significantly less active cells than inactive cells. Examples of sparse data are cloud points from LiDAR, or tiles extracted from WSI. With their own structure, sparse data have decreased memory footprint over non-sparse (*i.e.* dense) data such as images. Several

convolutional implementations designed for sparse input data have been proposed (Graham, 2015; Graham and van der Maaten, 2017; Riegler et al., 2017) revolving around the idea that convolutions should only be performed on active cells, which therefore decreases the number of computations by ignoring input regions with only inactive cells. In our work, we integrate sparse-input CNN within the MIL paradigm to shift the MIL paradigm towards a sparse convolutional one. From another point of view, we design a pooling layer that embed the MIL paradigm into a sparse fully convolutional classification architecture.

## 3. Methods

In this section, we present the processing of a WSI by SparseConvMIL, as illustrated in Figure 1 containing a first step of tile embeddings, followed by SparseConvMIL specific steps including (i) the sparse map construction, (ii) the sparse-input CNN processing of the sparse map, and (iii) specific data augmentation.

We consider a WSI $x \in \mathbb{R}^{3 \times w \times h}$ (3 channels, width $w$, height $h$) and a set of $K$ patches $(x_1, \ldots, x_K)$ extracted from $x$. A generic tile embedder $f_{\theta_1}$ (e.g. a ResNet architecture (He et al., 2016)) concurrently and independently computes the tiles embeddings $(f_{\theta_1}(x_1), \ldots, f_{\theta_1}(x_K))$.

**Sparse map construction** For each tile $x_k$, we also consider its location within the WSI denoted $(i_k, j_k)$, such as its center. A sparse embedding map $S^x$ is built by assigning each tile embedding to their associated tile location, while other locations are set inactive. Alternatively, $S^x$ can be formalized only by its active cells:

$$S^x = \left\{ \left[(i_k, j_k), f_{\theta_1}(x_{i_k,j_k})\right]; 1 \leq k \leq K \right\} \subset (\mathbb{N} \times \mathbb{N} \times \mathbb{H})^K \qquad (2)$$

In theory, a sparse map thus built has the same size as the input WSI. However in practice, few tiles can be extracted from a WSI *i.e.* the sparse map is very sparse. This implies that most convolutional operations would involve at most one active cell — essentially not leveraging the locality of the filters of the CNN. To address this issue, we introduce an additional parameter called the *downsampling* factor, noted $d$. Rather than assigning a tile $x_k$ at its associated location $(i_k, j_k)$ within $S^x$, it is assigned to locations $\left( \left\lfloor \frac{i_k}{d} \right\rfloor, \left\lfloor \frac{j_k}{d} \right\rfloor \right)$. $d$ sufficiently high would ensure that later neurons have at least two active cells in their receptive fields. We evaluate the impact of the downsampling factor in the experiments.

Because the sparse map construction only assigns vectors to locations, it is differentiable. In particular, an error signal from backpropagation can be assigned to vectors based on their locations — and subsequently to update the parameters of $f_{\theta_1}$. Once $S^x$ is built, it is forward into a sparse-input CNN.

**Sparse-input convolutional neural network** Sparse-input CNNs are essentially CNNs that operate specifically on sparse data, with reduced FLOPs and memory footprint. In particular with the formulation of Graham and van der Maaten (2017), a sparse-input convolution produces at most the number of input map active cells, by setting an inactive cell in the output for each input inactive cell. Mathematically, given a convolutional layer $U \in \mathbb{R}^{2f+1} \times \mathbb{R}^{2f+1} \times \mathbb{R}^o$ with a filter of half-size $f$, stride $s$, and $o$ output channels, convolving

$U$ on a sparse map $S^x$ produces the feature map $U \circledast S^x$ such that:

$$\forall i,j : (U \circledast S^x)_{i,j} = \begin{cases} \sum_{\substack{m=-f \\ n=-f}}^{f} U_{m+f,n+f} \cdot S^x_{si+m,sj+n} & \text{if } S^x_{si,sj} \neq \varnothing \\ \varnothing & \text{otherwise} \end{cases} \tag{3}$$

where $\varnothing$ indicates inactive cells and an inactive cell has no impact on the sum.

Similarly, for a pooling function $p$ such as max or average, the output of a $p$-pooling layer with filter size f and stride s on $S^x$ produces $p(S^x)$ such that:

$$\forall i,j : p(S^x)_{i,j} = \begin{cases} p\big(\{S^x_{si+m,sj+n}; 0 \leq m,n < f, S^x_{si+m,sj+n} \neq \varnothing\}\big) & \text{if } S^x_{si,sj} \neq \varnothing \\ \varnothing & \text{otherwise} \end{cases} \tag{4}$$

In particular, sparse-input adaptive global pooling layers with output size $o$ are defined as sparse-input pooling layers of both stride and filter size $\lfloor \frac{w}{o} \rfloor$ for the width and $\lfloor \frac{h}{o} \rfloor$ for the height. Activation functions are processed in the same fashion with stride 1, filter size 1 (therefore only on active cells) and with functions such as ReLU, tanh, or sigmoid.

Eventually a bag embedding is obtained after a succession of sparse-input (strided) convolutions and activations. To ensure that the spatial dimension of the bag embedding does not depend on the size of the input whole slide image, a (sparse-)adaptive global pooling layer is used at the end of the sparse-input CNN MIL operator — effectively transforming the output sparse feature map into a dense one. Bag scores can then be computed with any type of classifier, including multi-layer perceptrons.

**Context-aware specific data augmentation** The proposed approach benefits from additional data augmentation strategies over permutation non-invariant pooling strategies, precisely because it treats instances as non i.i.d. Spatial augmentations (*e.g.* flipping, rotations, local shuffling or elastic deformations) performed on tiles locations, or equivalently on the sparse map, can help reduce the burden of overfitting by artificially increasing the input data to the pooling CNN. Besides, these augmentations can be performed after tile embedding inference, implying that multiple sparse map spatial augmentations can be done with a low additional memory footprint. Examples of data augmented sparse maps are shown in Figure 3.

## 4. Experimental validation

We have used two data examples to demonstrate the extreme potentials of our method: CRCHISTOPHENOTYPE (subsection 4.1) and THE CANCER GENOME ATLAS (TCGA) (subsection 4.2).

### 4.1 Classical MIL dataset

**Dataset** The CRCHISTOPHENOTYPE (Sirinukunwattana et al., 2016) dataset consists of 100 haematoxylin and eosin-stained (H&E) $500 \times 500$ pixels histology images of colorectal adenocarcinomas. A total of 22,444 nuclei are annotated with (i) the position of their center and (ii) their class type *i.e.* one of epithelial, inflammatory, fibroblast, or miscellaneous.

We considered the binary task of classifying power fields as having epithelial cells or not: accurately detecting epithelial cells is a valuable clinical task since most cancers arise from the epithelium (Thiery et al., 2009). To do so, 27 pixels-wide images were extracted from all annotated cell centers. Although each 27×27 image has a binary annotation (epithelial or not), those were hidden during training, and only the power field-level labels were made available, which were set to 1 if at least one 27×27 image is epithelial. This resulted in 51 positive and 49 negative bags.

**Implementation details** SparseConvMIL was benchmarked along attention-based MIL approaches (Ilse et al., 2018), as well as instance and embedding-based max and mean pooling. All of these approaches shared the same training parameters which are detailed in Appendix B. This included (i) architectures of the tile embedding network $f_{\theta_1}$ and classifier $h_{\theta_3}$, (ii) hyper-parameters such as learning rate, optimizer, and batch size, and (iii) data augmentation. We now detail the specificities of each approach. Attention-based approaches used a two-layer neural network for attention with 128 hidden neurons as used in Ilse et al. (2018), resulting in 66304 pooling parameters and 131968 for the gated version. The proposed SparseConvMIL was implemented with two 12-channels convolutional filters with filter size 3 and stride 1, and ReLU activation, resulting in a module with 56628 parameters. For SparseConvMIL, the position of the center of each tile was used to build the sparse maps with a downsampling factor of 5, resulting in sparse maps of size $50 \times 50$.

| METHOD | ACCURACY | PRECISION | RECALL | F1-SCORE | AUC |
|---|---|---|---|---|---|
| Instance+max | 0.842±0.021 | 0.866±0.017 | 0.816±0.031 | 0.839±0.023 | 0 .914±0.010 |
| Instance+mean | 0.772±0.012 | 0.821±0.011 | 0.710±0.031 | 0.759±0.017 | 0 .866±0.008 |
| max | 0.824±0.015 | 0.884±0.014 | 0.753±0.020 | 0.813±0.017 | 0 .918±0.010 |
| mean | 0.860±0.014 | 0.911±0.011 | 0.804±0.027 | 0.853±0.016 | 0 .940±0.010 |
| Attention | 0.904±0.011 | **0.953**±0.014 | 0.855±0.017 | 0.901±0.011 | **0.968**±0.009 |
| Gated-Attention | 0.898±0.020 | 0.944±0.016 | 0.851±0.035 | 0.893±0.022 | **0.968**±0.010 |
| Proposed | **0.944**±0.019 | 0.929±0.021 | **0.944**±0.019 | **0.932**±0.024 | 0.958±0.008 |

Table 1: Results on CRCHISTOPHENOTYPE in mean ± standard deviation of 5 runs. Attention and Gated-Attention are from Ilse et al. (2018).

**Results** Results are reported in Table 1. The proposed approach achieved the best performance in terms of balanced accuracy and f1-score. Although its precision was slightly lower than the attention-based methods, it achieved a significantly higher recall, which is desirable for clinical considerations in order not to miss potentially arising tumor tissue.

### 4.2 Large-scale whole slide image dataset

**Dataset** 10000 whole slide images were downloaded from THE CANCER GENOME ATLAS from 32 cancer subtypes as detailed in Table 5, for a total of 5.57 TB of data. The only inclusive criteria was that WSI must display tumor material since the downstream task was cancer subtype classification which cannot be accurately done on benign samples. All WSI were tiled into 512 pixel-wide tiles with 128 pixels overlap on both sides at 10× magnification using the repository from Lerousseau et al. (2020). The cohort was split on a patient-basis in 4397, 2001 and 3602 slides for respectively the training, validation and testing sets.

**Implementation details** 14 MIL approaches were benchmarked: (i) embedding and instance-based meax, mean, and log-sum-exp, (ii) several flavors of attention-based approaches, (iii) a graph-CNN approach (Tu et al., 2019), and (iv) several flavors of SparseConvMIL. Similar to the previous experiment, all approaches share the same training context detailed in Appendix B including hyper-parameters, data augmentation, architectures of both tile embedder and classifier from WSI embeddings. For specific details of the benchmarked approaches, SparseConvMIL used a downsampling of 128. We experimented with only 2 convolutional layers with 32, or 128 channels. The attention module of attention MIL approaches were made of a 1 hidden layer perceptron with 128, 512 or 2048 neurons. Graph-based MIL was implemented with the same parameters as in Tu et al. (2019).

| METHOD | #PARAMS | ACCURACY | PRECISION | F1-SCORE | AUC | CE↓ |
|---|---|---|---|---|---|---|
| Random performance | N/A | 0.031 | 0.031 | 0.031 | 0.500 | 3.506 |
| Instance+max[†] | 0 | 0.417 | 0.365 | 0.360 | 0.879 | 2.027 |
| Instance+mean[†] | 0 | 0.463 | 0.417 | 0.414 | 0.905 | 1.783 |
| Instance+LSE[†] | 0 | 0.451 | 0.406 | 0.403 | 0.898 | 1.819 |
| max[†] | 0 | 0.441 | 0.434 | 0.403 | 0.913 | 1.821 |
| mean[†] | 0 | 0.488 | 0.463 | 0.456 | 0.917 | 1.604 |
| Attention-128 | 12k | 0.481 | 0.448 | 0.449 | 0.913 | 1.619 |
| Attention-512 | 219k | 0.487 | 0.451 | 0.453 | 0.912 | 1.616 |
| Attention-2048 | 985k | 0.472 | 0.452 | 0.452 | 0.909 | 1.621 |
| Gated-Attention-128 | 24k | 0.492 | 0.452 | 0.456 | 0.916 | 1.613 |
| Gated-Attention-512 | 261k | 0.487 | 0.447 | 0.450 | 0.911 | 1.629 |
| Gated-Attention-2048 | 1986k | 0.483 | 0.457 | 0.459 | 0.911 | 1.604 |
| Graph-CNN | 719k | 0.464 | 0.439 | 0.436 | 0.907 | 1.673 |
| Proposed-c32,c32 | 87k | **0.523** | **0.508** | **0.504** | **0.935** | **1.386** |
| Proposed-c128,c128 | 672k | **0.568** | **0.568** | **0.553** | **0.944** | **1.267** |
| Random performance | N/A | 0.031 | 0.031 | 0.031 | 0.500 | 3.506 |

Table 2: Results of the 32 classes classification on the TCGA dataset. PARAMS are the number of pooling parameters (in thousand) *i.e.* without considering tile embedder and classifier. ACCURACY is balanced. CE stands for cross-entropy. Random is the random performance. [†] denotes pooling methods that are non-parametric *i.e.* that cannot have parameters in their pooling operator.

**Results** Table 2 reports results for several metrics computed by averaging one-vs-all metrics for each class. In particular, the random performance is 0.031 for both accuracy, precision and f1-score, 0.5 AUC, and 3.506 cross-entropy. Although the number of pooling parameters differed for some methods, the comparisons are otherwise fair since both tile embedder and finale classifier were the same.

The proposed approach achieved superior results in all metrics and for all configurations. Even the smaller SparseConvMIL configuration outcompeted other benchmarked approaches with 0.568 of precision and balanced accuracy, f1-score of 0.553, and AUC of 0.944. This is extremely encouraging given that the task has 32 classes and that many classes are under-represented: for instance, 14 classes have less than 100 training samples (Table 6). Furthermore, SparseConvMIL seemed to scale better with parameters than

attention-based MIL. Indeed, increasing the parameters of SparseConvMIL significantly improved its performance whereas attention-based MIL stagnated with additional parameters. The graph-based approach notoriously underperformed, which may be due to the difficulty in choosing appropriate parameters for the many modules involved in this approach.

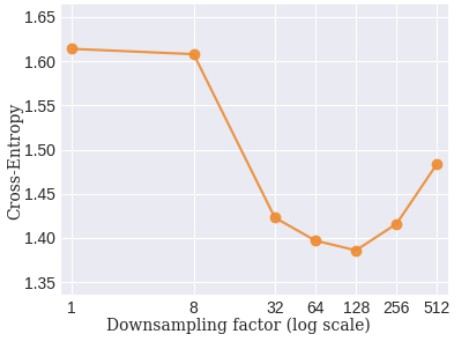

(a) varying the downsampling factor

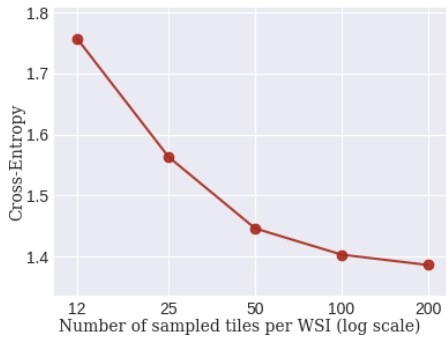

(b) varying the number of sampled tiles

Figure 2: Performance of SparseConvMIL-c32,c32 by varying the input sparsity on the TCGA experiment.

We aimed at understanding the limitations of the proposed approach. Figure 2a plots the performance of SparseConvMIL-c32,c32 on the same task but with various downsampling factors. In particular, the method performed significantly worse for low downsampling, which we conjecture is due to the fact that tiles are too far apart to exploit spatial context with convolutions. Furthermore, the performance of SparseConvMIL decreased for high downsampling, *e.g.* 256 or 512, which is probably due to uncoalesced sparse maps, where there may be duplicate coordinates for several tiles provoking a loss of input tiles. Meanwhile, Figure 2b plots the performance of SparseConvMIL by varying the number of tiles sampled per WSI for a downsampling of 128. Its performance increased with respect to the number of sampled tiles, which is not surprising since more increasing the number of sampled tiles provide additional information about the underlying whole slide images.

## 5. Conclusion

In this paper, we proposed a flexible and powerful sparse-input convolutional multiple instance learning approach for classifying whole slide images. SparseConvMIL demonstrated significantly better performance for the pan-cancer subtype classification of whole slide image, an extremely pertinent task for clinical purposes. Although some limitations of our approach have been highlighted, we believe that SparseConvMIL has the potential to become a gold standard for WSI classification and tile representation learning.

Our most important future work is to integrate interpretability through visualization, and notably by automatically extracting key instances. We obtained encouraging early results by using several common CNN visualization techniques such as Class Activation Mapping (Zhou et al., 2016), GradCAM Selvaraju et al. (2017), and DeepLIFT (Shrikumar et al., 2017). Meanwhile, there are endless possibilities on the choice of architectures for the sparse-input CNN part of our approach which can be investigated.

## Appendix A. Mathematical framework of multiple instance learning

Let us consider a set $X$ of bags (WSIs) $(x_i)_{1 \leq i \leq n}$ such that each bag $x_i$ is constituted of a set of $k_i$ instances (tiles) $\{x_{i,1}, x_{i,2}, \cdots, x_{i,k_i}\}$ where instances are from a domain $\mathcal{D}$. In particular, bags can have variables number of instances, or can share the same number of instances and, in that case, $\forall i, j, k_i = k_j$.

In its most general formulation, a MIL model $m$ can be written as a combination of 3 modules:

1. An instance-embedder $f_{\theta_1} : \mathcal{D} \to \mathcal{E}$ embedding instances into a space $\mathcal{E}$

2. A pooling operator $g_{\theta_2} : \prod \mathcal{E} \to \mathcal{F}$ processing a set (of arbitrary size) of instance embeddings into a bag embedding

3. A bag classifier $h_{\theta_3} : \mathcal{F} \to \mathcal{Y}$ projecting a bag embedding

such that

$$\forall x_i \in X, m(x_i) = g_{\theta_3}\Big(h_{\theta_2}\big(f_{\theta_1}(x_{i,1}), f_{\theta_1}(x_{i,2}), \cdots, f_{\theta_1}(x_{i,k_i})\big)\Big) \in \mathcal{Y}$$

Examples of pooling functions are:

$$\text{mean} \qquad : x_i \mapsto \frac{1}{k_i}\sum_{k=1}^{k_i} x_{i,k}$$

$$\text{max} \qquad : x_i \mapsto \max\{x_1, \ldots, x_{k_i}\}$$

$$\text{log-sum-exp (Ramon and De Raedt, 2000)} \quad : x_i \mapsto \frac{1}{M}\log\Big(\sum_{k=1}^{k_i}\exp(M \times x_{i,k})\Big)$$

$$\text{attention (Ilse et al., 2018)} \qquad : x_i \mapsto \sum_{k=1}^{k_i} \frac{\exp\big(w^\top \tanh(V x_{i,k}^\top)\big)}{\sum_{j=1}^{k_i}\exp\big(w^\top \tanh(V x_j^\top)\big)} \cdot x_{i,k}$$

$$\text{gated-attention (Ilse et al., 2018)} \qquad : x_i \mapsto \sum_{k=1}^{k_i} \frac{\exp\big(w^\top (\tanh(V x_{i,k}^\top) \odot \mathrm{sigm}(U x_{i,k}^\top))\big)}{\sum_{j=1}^{k_i}\exp\big(w^\top (\tanh(V x_j^\top) \odot \mathrm{sigm}(U x_j^\top))\big)} \cdot x_{i,k}$$

where $a \in \mathbb{N}^*$, $r \in \mathbb{R}^*$, $M \in \mathbb{R}^+$, $V \in \mathbb{R}^{L \times \dim(\mathbb{H})}$, $U \in \mathbb{R}^{L \times \dim(\mathbb{H})}$, $w \in \mathbb{R}^{L \times 1}$, $L \in \mathbb{N}^*$ are parameters, $\odot$ is the elementwise multiplication, and sigm is the elementwise sigmoid function. The max operator can also be substituted or combined with the min operator. The log-sum-exp is also known as the softplus function and is considered as a smooth approximation to the max function. Attention-based approaches (Ilse et al., 2018) leverage an attention module formalized with a one hidden layer perceptron, that computes one score per input instance embedding which are then normalized such that they sum to 1, as to accommodate with a potentially varying number of instances. All of these functions output a vector (or bag embedding) with the same shape as the $k_i$ input vectors. It is possible to combine them in many ways such as to obtain output vectors of higher dimensions *e.g.* by using concatenation, summation, average or sequential combinations of themselves. These operators can be used for instance-based or embedding-based multple instance learning.

# Appendix B. Implementation details of the experimental validation

## B.1 Epithelial classification on CRCHistoPhenotype

All methods shared the same training parameters as follows:

- The instance embedding model $f_{\theta_1}$ (Table 3) proposed in Sirinukunwattana et al. (2016) and used in Ilse et al. (2018) was employed.

- Loss function was binary cross-entropy

- Optimizer was the Adam (Kingma and Ba, 2014) with default momentum values $\beta$ of 0.9 and 0.999, learning rate of $1e^{-4}$, weight decay of $5e^{-4}$, batch size of 1 for 100 epochs.

- Data augmentation consisted in the next functions in that order:

  1. Random vertical and horizontal flip.

  2. Random rotation.

  3. H&E color augmentation (Ruifrok et al., 2001): H&E histopathological slides are originally uncoloured. The two stains Haematoxylin and Eosin are applied which respectively color nuclei and cytoplasm. Therefore, the true color space of H&E slides is made of the two vectors H and E rather than R, G, and B. Each tile was deconvoluted in the HE space using scikit-learn (Pedregosa et al., 2011) v0.24.2 (behavior changes depending on the version for the considered functions) with H vector value of $H = [0.650, 0.704, 0.286]^\top$ and E value of $E = [0.071, 0.994, 0.112]^\top$. Then, two independent random gaussian variables with mean 1 and standard deviation of 3 were sampled, and multiplied to $H$ and $E$. These product of these multiplications were used to convert the tile from the (H, E, residual) space back to the RGB space.

  4. Random crop of a 128 pixel-wide region.

  5. Channel-wise standard scaling with RGB mean and standard deviation extracted from the training set.

| Layer ID | Layer type | Layer parameters |
|---|---|---|
| 1 | Conv | Filter width 4, stride 1, padding 0, ReLU |
| 2 | Maxpool | Filter width 2, stride 2 |
| 3 | Conv | Filter width 3, stride 1, padding 0, ReLU |
| 4 | Maxpool | Filter width 2, stride 2 |
| 5 | Fully connected | 512 neurons, ReLU |
| 6 | Dropout | 0.25 |
| 7 | Fully connected | 512 neurons, ReLU |
| 8 | Dropout | 0.25 |

Table 3: Tile embedding model $f_{\theta_1}$ from Sirinukunwattana et al. (2016) used in the CRCHISTOPHENOTYPE experiment.

| Layer ID | Layer type | Layer parameters |
|---|---|---|
| 1 | Conv | Filter width 4, stride 1, padding 0, ReLU |
| 2 | Maxpool | Filter width 2, stride 2 |
| 3 | Conv | Filter width 3, stride 1, padding 0, ReLU |
| 4 | Maxpool | Filter width 2, stride 2 |
| 5 | Fully connected | 512 neurons, ReLU |
| 6 | Dropout | 0.25 |
| 7 | Fully connected | 512 neurons, ReLU |
| 8 | Dropout | 0.25 |
| 9 | max or | |
| 9 | mean or | |
| 9 | attention module or | |
| 9 | Sparse-input CNN | |
| 10 | Fully connected | 1 output neuron, Sigmoid |
| Layer ID | Layer type | Layer parameters |
| 1 | Conv | Filter width 4, stride 1, padding 0, ReLU |
| 2 | Maxpool | Filter width 2, stride 2 |
| 3 | Conv | Filter width 3, stride 1, padding 0, ReLU |
| 4 | Maxpool | Filter width 2, stride 2 |
| 5 | Fully connected | 512 neurons, ReLU |
| 6 | Dropout | 0.25 |
| 7 | Fully connected | 512 neurons, ReLU |
| 8 | Dropout | 0.25 |
| 9 | Fully connected | 1 output neuron, Sigmoid |
| 10 | Max-MIL/Mean-MIL | |

Table 4: Complete models from the CRCHISTOPHENOTYPE experiment. The top table displays architectures for embedding-level approaches, while the bottom row displays architectures for instance-level approaches.

For SparseConvMIL, the position of the center of each tile was used to build the sparse maps before applying spatial data augmentation consisting of random flips, rotations, and per axis scaling as detailed in section B. The sparse-input CNN was made of 2 convolutional layers of 12 channels, filter size 3, stride 1, activated with ReLU. An adaptive global average pooling layer converted the second layer sparse signal into a dense signal. The implementations of other methods are detailed in Table 4 and (Ilse et al., 2018).

All approaches are trained end-to-end. The 100 power fields were split into 55, 20, 24 samples for respectively the training, validation and testing set. The validation set is used to select the snapshot with least validation error for inference on the testing set. The training/testing process was performed 5 times for each method to derive confidence intervals.

| Project ID | Description | Location | # WSI |
|---|---|---|---|
| TCGA-ACC | Adrenocortical carcinoma | Adrenal gland | 96 |
| TCGA-BLCA | Bladder Urothelial Carcinoma | Bladder | 298 |
| TCGA-BRCA | Brain Lower Grade Glioma | Breast | 1052 |
| TCGA-CESC | Breast invasive carcinoma | Cervix | 190 |
| TCGA-CHOL | Cervical squamous cell carcinoma and endocervical adenocarcinoma | Bile ducts | 46 |
| TCGA-COAD | Cholangiocarcinoma | Colon | 508 |
| TCGA-DLBC | Colon adenocarcinoma | Lymph nodes | 39 |
| TCGA-ESCA | Esophageal carcinoma | Esophagus | 130 |
| TCGA-GBM | Glioblastoma multiforme | Brain | 647 |
| TCGA-HNSC | Head and Neck squamous cell carcinoma | Head and Neck | 412 |
| TCGA-KICH | Kidney Chromophobe | Kidney | 104 |
| TCGA-KIRC | Kidney renal clear cell carcinoma | Kidney | 773 |
| TCGA-KIRP | Kidney renal papillary cell carcinoma | Kidney | 242 |
| TCGA-LGG | Liver hepatocellular carcinoma | Brain | 509 |
| TCGA-LIHC | Lung adenocarcinoma | Liver | 287 |
| TCGA-LUAD | Lung squamous cell carcinoma | Lung | 514 |
| TCGA-LUSC | Lymphoid Neoplasm Diffuse Large B-cell Lymphoma | Lung | 511 |
| TCGA-MESO | Mesothelioma | Mesothelium | 55 |
| TCGA-OV | Ovarian serous cystadenocarcinoma | Ovary | 477 |
| TCGA-PAAD | Pancreatic adenocarcinoma | Pancreas | 147 |
| TCGA-PCPG | Pheochromocytoma and Paraganglioma | Adrenal gland | 132 |
| TCGA-PRAD | Prostate adenocarcinoma | Prostate | 426 |
| TCGA-READ | Rectum adenocarcinoma | Rectum | 180 |
| TCGA-SARC | Sarcoma | Soft tissues | 292 |
| TCGA-SKCM | Skin Cutaneous Melanoma | Skin | 336 |
| TCGA-STAD | Stomach adenocarcinoma | Stomach | 383 |
| TCGA-TGCT | Testicular Germ Cell Tumors | Testicular | 138 |
| TCGA-THCA | Thymoma | Thyroid | 384 |
| TCGA-THYM | Thyroid carcinoma | Thymus | 105 |
| TCGA-UCEC | Uterine Carcinosarcoma | Uterus | 465 |
| TCGA-UCS | Uterine Corpus Endometrial Carcinoma | Uterus | 60 |
| TCGA-UVM | Uveal Melanoma | Skin | 62 |
| Total | Vitually all solid cancer subtypes | Pan-location | 10000 |

Table 5: Distribution of cancer subtypes (classes), locations, and number of WSI in the total cohort of 10000 slides involved in our experiments. The first column indicates the official TCGA project ids which groups all cases from the same cancer subtype. The second columns shows the location of each cancer subtype. The third displays the total number of WSI for each cancer subtype. The last line indicates the total of the cohort.

## B.2 Subtype classification on The Cancer Genome Atlas

All benchmarked methods used a ResNet34 architecture He et al. (2016) pre-trained on Imagenet Deng et al. (2009) as the instance embedding $f_{\theta_1}$. To obtain embeddings instead of the probabilities output of ResNet34, the last classifier layer was removed, resulting in 512 output channels per tile instead of 1 probability. All benchmarked approaches shared the same MIL classifier $h_{\theta_3}$ which was made of one 512-neurons ReLU activated fully connected layer followed by a 32-output fully connected layer (there are 32 classes). During training, 200 randomly cropped $128 \times 128$ pixel tiles were randomly sampled within each WSI. Hyperparameters were shared across all benchmarked approaches and were:

- Loss function was binary cross-entropy.

- Optimizer was the Adaptive Momentum (Kingma and Ba, 2014) with default momentum values, learning rate of $1e^{-4}$, weight decay of $1e^{-4}$, batch size of 10 (or 2000 taking into account the number of tiles per WSI) for 200 epochs.

- Due to significant imbalance in the class distribution, oversampling was employed during training with frequencies equal to the inverse of the class counts.

- Data augmentation was the same as in the CRCHistoPhenotype experiment (see subsection B.1).

| Cancer subtype (class) | # training WSI |
|---|---|
| lymphoid neoplasm diffuse large b-cell lymphoma | 17 |
| cholangiocarcinoma | 20 |
| mesothelioma | 24 |
| uterine carcinosarcoma | 26 |
| uveal melanoma | 27 |
| adrenocortical carcinoma | 42 |
| kidney chromophobe | 46 |
| thymoma | 46 |
| esophageal carcinoma | 57 |
| pheochromocytoma and paraganglioma | 58 |
| testicular germ cell tumors | 61 |
| pancreatic adenocarcinoma | 65 |
| rectum adenocarcinoma | 79 |
| cervical squamous cell carcinoma and endocervical adenocarcinoma | 83 |

Table 6: Number of training whole slide images for the 14 cancer subtypes (classes) with less than 100 training samples. This table illustrates that some classes are heavily underrepresented in the training set, which can challenge the accurate and efficient learning of features discriminative for subtype classification.

Additionally, SparseConvMIL and the graph-based approaches has the following data augmentation directly performed on tiles coordinates as follows (no effect on other approaches):

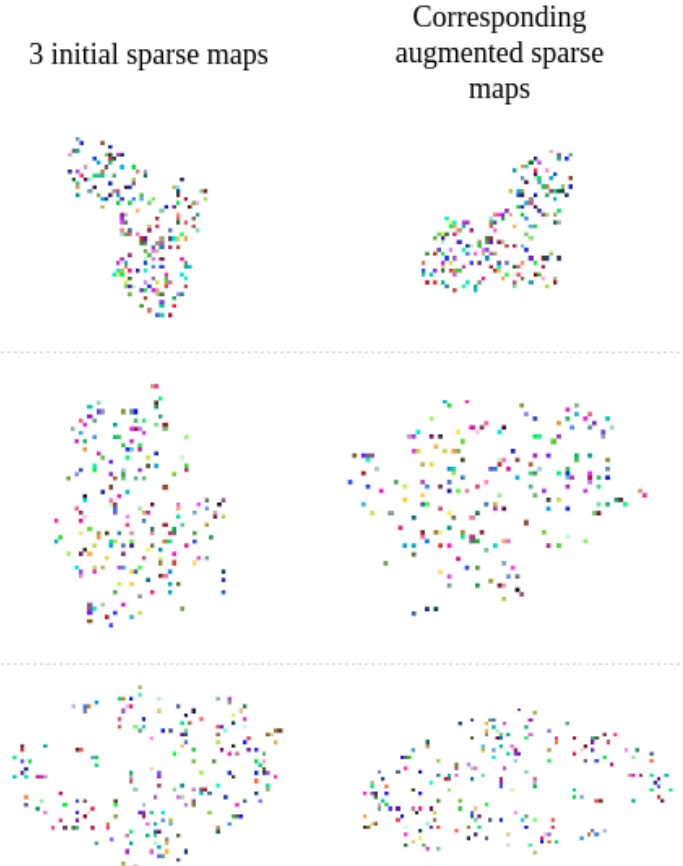

Figure 3: Illustration of SparseConvMIL specific data augmentation. 3 sparse maps are represented in the first column, 1 per line. For each sparse map, 300 512 pixel wide tiles were randomly sampled from the tissue section of WSI, and were spatially represented as coordinates within sparse maps. Each sparse map was spatially data augmented with random flips, rotations, scaling per axis and is displayed in the second column. Color for tiles is used for tracking purposes.

- random vertical and horizontal coordinates flips

- random coordinates rotation with an angle uniformly sampled within $[0, 2\pi]$

- random zoom for both x and y axes by sampling one value per axis in range $[0.7, 1.3]$

All approaches were trained end-to-end. For each method, the training process lasted approximately 1 week on 2 Nvidia V100. For fairness of comparisons, all approaches shared the same tile embedding function and classifier function: the only varying method was the pooling operator which can scale with the number of parameters for attention-based, graph-based and sparse-convolutional-based approaches but not for non-parametric approaches of max, mean and LSE.

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
