# OpenReview forum: "SparseConvMIL: Sparse Convolutional Context-Aware Multiple Instance Learning for Whole Slide Image Classification"
_MICCAI.org/2021/Workshop/COMPAY — COMPAY 2021_

### Official Review · Reviewer_pw5L · 2021-08-08
**The authors proposed a novel method for context-aware multi-instance learning (MIL). Unlike traditional MIL methods, they develop a permutation variant pooling method to capture the spatial context in whole slide images. Their proposed method has shown superior performance as compared to its counterparts on two different histology classification tasks.**

**Rating:** 7
**Confidence:** 5

**Review:**

In a binary classification task, AUC is often preferred over accuracy. Although the proposed method shows superior performance in terms of accuracy on the CRCHistoPhenotyp dataset, the AUC score of the proposed method is still low as compared to Attention and Gated-Attention based methods. Therefore, limiting the significance of the proposed method over its counterparts.

For TCGA 32-class classification, the proposed achieved a higher AUC score in a one-vs-rest setting. However, it would be interesting to know what is the performance of the proposed method in a one-vs-one setting especially for classes with less than 100 samples.

The order of tables and figures in the manuscript is quite unusual as the tables and figures referred earlier were appeared later in the manuscript and vice versa.

---

### Official Review · Reviewer_uuUy · 2021-08-09
**SparseConvMIL - good paper!**

**Rating:** 9
**Confidence:** 5

**Review:**

The authors propose a method to represent WSIs as a sparse map of tile embeddings. Using this sparse map, a sparse CNN is utilised to make a WSI-level (or image-level) prediction and outperforms recently used MIL techniques. Utilising a sparse map in this way enables the spatial relationship between embeddings to be preserved.

The introduction to the problem is very well motivated and the challenge of dealing with large-scale WSIs is clearly stated. The paper is clearly written, with sufficient experiments to prove the merit of the approach.

Figure 2b is interesting as it shows the performance with increasing number of samples. However, do you think that these tiles could be more intelligently sampled? Randomly sampling may miss important cues within the image. For example, it may struggle identifying isolated tumour cells for the task of WSI tumour classification. Also, I could not find in the text the actual number of sampled tiles used to populate the results in the tables. Also, the number of optimal sampled tiles may differ depending on the task. Could the number of sampled tiles be learned by the network?

The use of augmentation on the sparse maps is interesting. Please can you include an ablation without the use of this augmentation.

Overall, the paper is a novel and interesting new way of dealing with WSIs for classification tasks and think it will be a useful new addition to the field of computational pathology.

Minor notes:
- 10000 - insert comma to make it 10,000
- twofold -> two fold
- Bottom of page 3 - cloud points or point clouds?

---

### Decision · Program_Chairs · 2021-08-25

Accept